# WATCH LESS, DO MORE: IMPLICIT SKILL DISCOVERY FOR VIDEO-CONDITIONED POLICY

**Jiangxing Wang**
School of Computer Science
Peking University
jiangxiw@stu.pku.edu.cn

**Zongqing Lu**[*]
School of Computer Science
Peking University, BAAI
zongqing.lu@pku.edu.cn

## ABSTRACT

In this paper, we study the problem of video-conditioned policy learning. While previous works mostly focus on learning policies that perform a single skill specified by the given video, we take a step further and aim to learn a policy that can perform multiple skills according to the given video, and generalize to unseen videos by recombining these skills. To solve this problem, we propose our algorithm, **Watch-Less-Do-More**, an information bottleneck-based imitation learning framework for implicit skill discovery and video-conditioned policy learning. In our method, an information bottleneck objective is employed to control the information contained in the video representation, ensuring that it only encodes information relevant to the current skill (**Watch-Less**). By discovering potential skills from training videos, the learned policy is able to recombine them and generalize to unseen videos to achieve compositional generalization (**Do-More**). To evaluate our method, we perform extensive experiments in various environments and show that our algorithm substantially outperforms baselines (up to **2x**) in terms of compositional generalization ability.

## 1 INTRODUCTION

As large language models (LLMs) have demonstrated remarkable zero-shot and few-shot generalization abilities (Brown et al., 2020; Ouyang et al., 2022), the research focus of decision-making policies has also shifted from addressing a specific task, such as mastering an environment via reinforcement learning (Sutton, 2018) or replicating a dataset via imitation learning (Hussein et al., 2017), to completing diverse tasks based on given instructions. These instructions can be treated as goals for the decision-making models, and encompass modalities such as text (Nair et al., 2022; Carta et al., 2023), goal image (Yadav et al., 2023b;a), or future state (Cui et al., 2022; Lee et al., 2024). To achieve such goal-conditioned policies, a variety of methods have been proposed and achieved great success across multiple domains (Liu et al., 2022). However, the aforementioned goal specifications often overlook dynamic information, such as the ordering of task completion or the method of task completion (if there are many). In contrast, video offers a natural way to represent these details, thereby leading to a line of research exploring video-conditioned policy learning (Eze & Crick, 2024b).

Existing methods for video-conditioned policy learning have been applied to various scenarios, including robotic manipulation (Chane-Sane et al., 2023; Shin et al., 2023; Jiang et al., 2023), navigation (Zhou et al., 2024), open-world agent (Cai et al., 2024), and autonomous driving (Shin et al., 2024). Taking different videos as input, the learned policy can be deployed to perform different skills to solve corresponding tasks. However, these methods often consider only the video demonstration of a single task (Chane-Sane et al., 2023) and only the object-level generalization (Jiang et al., 2023). In real-world applications, we often want the learned policy to perform a set of different skills to achieve a combination of multiple tasks.

When the video demonstration of a task combination is given, an ideal policy should directly perform skills as demonstrated in the given video. To train such a policy, researchers have explored

---

[*]Corresponding author.

skill-based imitation learning methods (Xu et al., 2023; Wang et al., 2023; Shin et al., 2023; 2024). However, these methods often require explicit video segmentation annotations, or videos of another embodiment to train a skill-based policy, which greatly increases the difficulty of the data collection process. Therefore, in this paper, we consider whether it is possible to learn a video-conditioned policy that can perform multiple skills without these requirements. Moreover, as we consider videos of different task combinations, we also expect the learned policy to achieve compositional generalization (Lin et al., 2023), that is, it can still perform well on task combinations that have not been during training.

To fulfill such an expectation, we propose our algorithm, **W**atch-**L**ess-**D**o-**M**ore (**WL-DM**), an information bottleneck-based imitation learning framework for implicit skill discovery and video-conditioned policy learning. For a video-conditioned policy, the given video can be considered as a sequence of tasks. As the policy can only work on one task at a time, it should be able to perform well by focusing only on the current task, instead of the entire task sequence. Based on this intuition, WL-DM employs the information bottleneck method (Tishby et al., 2000) to control the information contained in the video representation. This is accomplished by 1) minimizing the mutual information (Cover, 1999) between video and its representation to reduce the information contained in the video representation and 2) maximizing the mutual information between video representation and the current skill to preserve enough information related to the current task. To better understand the effect of this information bottleneck method, we further build a theoretical connection between the proposed method and the intuition behind it. Using this method, WL-DM makes the learned policy only to consider the current task, which achieves the *implicit* video segmentation without requiring explicit video segmentation annotations. The advantage of considering only the current task can be related to the compositional generalization ability. When an unseen video is given, the video-conditioned policy learned by WL-DM can implicitly decompose the unseen video into seen tasks, and perform corresponding skills, thus facilitating the compositional generalization ability of the learned video-conditioned policy. To further validate our algorithm, we propose a practical implementation of our method and conduct various empirical evaluations across diverse environments. The experimental results indicate that WL-DM achieves substantially better compositional generalization ability than baselines, demonstrating the effectiveness of our method.

Our contributions can be summarized as follows:

- We propose our method, Watch-Less-Do-More (WL-DM), an information bottleneck-based imitation learning framework for implicit skill discovery and video-conditioned policy learning, where two different mutual information terms work together to ensure the video representation contains only information related to the current task.

- The intuition behind WL-DM is that the optimal policy should behave similarly when conditioned on all tasks and when conditioned on only the current task. To better explain our method, we further build a theoretical connection between WL-DM and this intuition.

- We propose a practical implementation of our algorithm and perform empirical evaluations in Frank Kitchen (Gupta et al., 2020) and Meta world (Yu et al., 2020) to demonstrate the effectiveness of WL-DM. The experimental results indicate that WL-DM achieves (up to **2x**) better compositional generalization ability compared to baselines.

## 2 RELATED WORK

### 2.1 LEARNING FROM VIDEOS

Using massive Internet data to train language models has been proven to be successful and has resulted in a trend of research on large language models (Brown et al., 2020; Touvron et al., 2023). Inspired by this success, researchers have begun to pay attention to another type of data wildly available on the Internet, video data, and produced a series of studies on learning from videos (McCarthy et al., 2024; Eze & Crick, 2024a). For decision-making models, video data can be used in various ways, such as reward function learning (Escontrela et al., 2023; Sermanet et al., 2018; Chen et al., 2021a), dynamic model learning (Baker et al., 2022), representation learning (Nair et al., 2023), and policy learning (Jang et al., 2022; Jiang et al., 2023; Chane-Sane et al., 2023; Shin et al., 2023; 2024). Our paper belongs to the last category, that is, using video demonstrations as instructions to learn a video-conditioned policy. It is worth noting that previous work in this category often focuses

only on demonstration videos containing a single task (Chane-Sane et al., 2023), or requires aligned data of other modalities (Jang et al., 2022; Shin et al., 2023; 2024). This can be attributed to the lack of clear goal labels in demonstration videos (McCarthy et al., 2024). Therefore, when dealing with videos containing multiple tasks, we often need to introduce information in other modalities to provide segmentation annotations for the video, to distinguish the tasks to be completed at each stage (Shin et al., 2023; 2024). Unlike previous work, in this paper, we attempt to directly learn a video-conditioned policy capable of handling videos containing multiple tasks, without introducing additional segmentation annotations.

## 2.2 ONE-SHOT IMITATION LEARNING

One-shot imitation learning was originally introduced in Duan et al. (2017), where the goal of this problem is to learn a policy that can quickly adapt to a new task given a single demonstration. For one-shot imitation learning, we can achieve it through different learning methods such as meta-learning (Duan et al., 2017; Finn et al., 2017), semi-supervised learning (Wu et al., 2024), and imitation learning (Jang et al., 2022; Cui et al., 2022; Jiang et al., 2023). Specifically, our method falls into the last category: we assume the existence of an imitation learning dataset paired with video demonstrations, such that we can use this dataset to train a video-conditioned policy.

One-shot demonstrations can be presented in various formats, such as trajectories (Cui et al., 2022; Lee et al., 2024), videos (Dasari & Gupta, 2021; Jain et al., 2024; Wang et al., 2023; Xu et al., 2023; Chane-Sane et al., 2023), multimodal information (Jiang et al., 2023; Shin et al., 2023; 2024), etc. In this paper, we consider adapting to new tasks through video demonstrations, that is, one-shot video imitation learning. In previous work, video demonstrations often only include a single task, and the adaptation to new tasks mainly focuses on differences at the embodiment and object level (locations, textures, etc.) (Dasari & Gupta, 2021; Mandi et al., 2022; Chane-Sane et al., 2023). Unlike these studies, we consider video demonstrations containing multiple tasks and focus on adaptation at the level of task combination. For this setting, previous work generally assumes the existence of data corresponding to another embodiment (Wang et al., 2023; Xu et al., 2023) or assumes information in other modalities to provide video segmentation annotations (Shin et al., 2023; 2024). Unlike these works, we do not assume additional data and learn a video-conditioned policy that can finish multiple tasks solely through the information contained in the videos.

## 2.3 COMPOSITIONAL GENERALIZATION

Due to the compositional nature of natural language, most previous work considers the compositional generalization problem over language instructions. For example, Oh et al. (2017) proposed a method based on hierarchical reinforcement learning that enables the policy to generalize to unseen command combinations and longer command sequences at test time. Stengel-Eskin et al. (2022) combined the transformer model and the masking mechanism to obtain generalization over object combinations. The attention mechanism for compositional generalization was further investigated by Spilsbury & Ilin (2022), and a method utilizing sparse factored attention for goal identification was proposed. Modular architecture is another way to induce compositional generalization. Carvalho et al. (2023) proposed modular successor features to enhance the compositional generalization ability, and Logeswaran et al. (2023) directly considered an additive decomposition of the state-action value function to obtain the generalization ability over language instructions.

Unlike these studies, we consider the generalization across different task combinations based on video demonstrations. During training, we only have access to a subset of task combinations and their corresponding video demonstrations. Our goal is to enable the policy to decompose different tasks from the videos and acquire skills to solve these tasks. At test time, the policy is expected to reproduce an unseen video demonstration by combining a set of skills learned in the training set. This setting has been studied by Wang et al. (2023); Xu et al. (2023); Shin et al. (2023; 2024). However, Wang et al. (2023); Xu et al. (2023) focused on the cross-embodiment scenario, thus requiring video data from another embodiment, and Shin et al. (2023; 2024) required language information to provide segmentation annotations for videos. Unlike them, our method incorporates an information bottleneck-based objective to achieve implicit video segmentation and skill discovery, without the need for other sources of information.

## 3 PROBLEM FORMULATION

In this paper, we consider the video-conditioned policy learning problem. This problem can be formulated as a special case of the goal-conditioned Markov Decision Process (MDP) (Nasiriany et al., 2019) and defined by a tuple $\langle \mathcal{S}, \mathcal{G}, \mathcal{A}, P, R, \rho_0, \gamma \rangle$. Similar to the general MDP, $\mathcal{S}$ is the set of states, $\mathcal{A}$ is the set of actions, $P(s_{t+1}|s_t, a_t)$ is the transition probability, $\rho_0$ is the initial state distribution and $\gamma$ is the discount factor. Additionally, we have $\mathcal{G}$ as the set of goals, which will also affect the reward function $R(s_t, a_t, g)$. For a goal-conditioned policy $\pi(a_t|s_t, g)$ with a given goal $g$, we want it to maximize the following objective:

$$\mathcal{J}(\pi) = \mathbb{E}_{s_0 \sim \rho_0, a \sim \pi, s' \sim P}[\sum_t \gamma^t r(s_t, a_t, g)].$$

As we focus on the video-conditioned policy learning, we assume our goals to be videos $\mathcal{G} = \mathcal{V}$, such that a goal-conditioned policy $\pi(a_t|s_t, g)$ becomes a video-conditioned policy $\pi(a_t|s_t, v)$. Moreover, we consider the case where each video $v = (\mathrm{k}_0, \cdots \mathrm{k}_N)$ contains multiple tasks $\mathrm{k} \in \mathcal{T}$, where $N$ is the number of tasks and $\mathcal{T}$ is the set of all possible tasks. To evaluate the compositional generalization ability of the video-conditioned policy, we assume two video sets $\mathcal{V}_{\text{train}}$ and $\mathcal{V}_{\text{test}}$, such that there is no overlapping between the train video set and test video set $\mathcal{V}_{\text{train}} \cap \mathcal{V}_{\text{test}} = \varnothing$ and both video sets contain all possible tasks $\bigcup_{v \in \mathcal{V}_{\text{train}}, \mathrm{k} \in v} \mathrm{k} = \bigcup_{v \in \mathcal{V}_{\text{test}}, \mathrm{k} \in v} \mathrm{k} = \mathcal{T}$. The video-conditioned policy will be trained in $\mathcal{V}_{\text{train}}$ to maximize $\mathbb{E}_{v \sim \mathcal{P}_{\text{train}}} \mathcal{J}(\pi)$ and will be tested in $\mathcal{V}_{\text{test}}$ in terms of $\mathbb{E}_{v \sim \mathcal{P}_{\text{test}}} \mathcal{J}(\pi)$, where $\mathcal{P}_{\text{train}}$ and $\mathcal{P}_{\text{test}}$ are uniform distributions across $\mathcal{V}_{\text{train}}$ and $\mathcal{V}_{\text{test}}$ respectively.

## 4 METHOD

In this section, we introduce our method, Watch-Less-Do-More (WL-DM). The intuition behind our method is that we want the video-conditioned policy to make decisions relying not on the entire video, but only on information related to the current task, thereby achieving implicit video segmentation and skill discovery. To achieve this, we propose an information bottleneck-based objective and theoretically establish the connection between this objective and our intuition. By decomposing training videos into a combination of different skills, the video-conditioned policy can handle unseen videos by recombining these skills to complete the required task combinations demonstrated in the unseen video.

### 4.1 INTUITION: FOCUSING ON THE CURRENT TASK

As formulated in Section 3, we assume that each video $v$ contains $N$ tasks $[\mathrm{k}_0, \cdots, \mathrm{k}_N]$ that need to be completed and the completion of these tasks is independent. In this case, we further assume a training set $\mathcal{D} = \{\tau_i, v_i\}$, where $\tau_i = (s_0, a_0^*, \cdots, s_T, a_T^*)_i$ is the expert trajectory corresponding to the video $v_i = (f_0, \cdots, f_T)$ and $f_i$ is the video frame at each timestep. Given such a dataset, we can easily learn a video-conditioned policy $\pi(a_t|s_t, v)$ through imitation learning (Hussein et al., 2017) that can complete different task combinations given different videos, at least within the coverage of the training set. For example, the policy can be trained via the following behavior-cloning loss:

$$\begin{aligned} \mathcal{L}_{\text{BC}}(\theta, \phi) &= -\mathbb{E}_{s_t, a_t^*, v \sim \mathcal{D}} \Big[ \log \pi(a_t^*|s_t, v) \Big] \\ &= -\mathbb{E}_{s_t, a_t^*, v \sim \mathcal{D}} \Big[ \log \mathrm{f}_\theta(a_t^*|s_t, \mathrm{g}_\phi(v)) \Big], \end{aligned} \tag{1}$$

where $\mathrm{g}_\phi$ is the video encoder and $\mathrm{f}_\theta$ is the action decoder.

A potential problem with this training method is that, when the size of our training set is limited, the learned policy can easily overfit (Ying, 2019) to videos in the training set. This problem causes the learned policy to focus too much on the details of these videos to distinguish them completely, while ignoring the fact that these videos are composed through elements of the same task set. In such a case, when an unseen video is given, i.e., an unseen combination of tasks, the performance of the learned policy may decrease dramatically due to the overfitting issue. To address this problem, we need to focus on the fact that all videos are composed through elements of the same task set $\mathcal{T}$. Even for those unseen videos, although the corresponding task combinations are not included in the

training set, each task that constitutes them has already been covered in the training set. Therefore, if we can decompose videos into individual tasks and train the policy based on the decomposed tasks, such that $\pi(a_t|s_t, v) = \pi(a_t|s_t, \text{v}_{\text{cur}})$, where $\text{v}_{\text{cur}}$ is the video segment corresponding to $\text{k}_{\text{cur}}$ and $v = (\text{v}_{\text{cur}}, \text{v}_{\text{other}})$, the policy can then handle unseen videos as all the skills demonstrated in the videos have been covered and trained in the training set, which is commonly known as compositional generalization (Lin et al., 2023). However, such a task-level video segmentation annotation could be inaccessible in many cases. In this paper, we do not assume this kind of annotations as in previous work (Shin et al., 2023; 2024). To achieve a similar effect, we propose an information bottleneck-based method, allowing the policy to implicitly decompose demonstration videos, enabling it to rely only on the information related to the current skill when making decisions, thereby achieving the compositional generalization ability.

## 4.2 INFORMATION BOTTLENECK FOR VIDEO-CONDITIONED POLICY LEARNING

As described in Section 4.1, when all tasks can be completed independently, a video-conditioned policy $\pi(a_t|s_t, v) = \pi(a_t|s_t, \text{v}_{\text{cur}})$ can achieve the compositional generalization ability. However, a video contains information about not only the current task $\text{k}_{\text{cur}}$ but also other tasks $\text{k}_{\text{other}}$. Therefore, we need an additional objective to train the video encoder $\text{g}_\phi$, such that it produces a similar representation for $v$ and $\text{v}_{\text{cur}}$, and we can then ensure $\pi(a_t|s_t, v) = \pi(a_t|s_t, \text{v}_{\text{cur}})$. To achieve this, we need to reduce the mutual information between video representations $h_v$ and the video segments of other tasks $\text{v}_{\text{other}}$, and the reason can be seen from the following theorem:

**Theorem 1.** *If we have* $\text{MI}(h_v; \text{v}_{\text{other}}|s, \text{v}_{\text{cur}}) = 0$*, then* $D_{\text{KL}}\big(\pi(a|s, v)||\pi(a|s, \text{v}_{\text{cur}})\big) = 0$ *for all state-video pairs* $(s, v) \in \mathcal{S} \times \mathcal{V}$ *with non-zero probability* $P(s, v) > 0$.

*Proof.* See Appendix B. □

This theorem suggests the necessity of reducing the information of other tasks contained in the video representation. However, since we do not assume any video segmentation annotation, we cannot directly obtain segments corresponding to the current task and other tasks from the video, and therefore cannot directly manipulate the mutual information. Hence, we use a constructive method to manipulate the information in the video representation indirectly. Specifically, we first minimize the mutual information between the video representation $h_v$ and the entire video $v$ to minimize the amount of information contained in the video representation. At the same time, we maximize the mutual information between the video representation and some approximation of the current skill (which will be discussed later). Since different skills are required for performing different tasks, we can in this way indirectly ensure that the video representation still retains a certain amount of information about the current task. Putting these two terms together, we can construct the following objective, which is often referred to as the information bottleneck (Tishby et al., 2000):

$$\mathcal{L}_{IB} = \text{MI}(h_v; v|s) - \alpha \, \text{MI}(h_v; z|s), \tag{2}$$

where $z$ is an approximated representation of the current skill, and $\alpha$ is the coefficient for the trade-off between two mutual information terms. As discussed in Tishby & Zaslavsky (2015), the information bottleneck is often used to learn a compact representation, which in our case is to dismiss the irrelevant part $\text{k}_{\text{other}}$ and retain the relevant part $\text{k}_{\text{cur}}$. In the following two sections, we discuss how to compute this objective in practice.

## 4.3 MINIMIZING MUTUAL INFORMATION WITH VIDEO

The first term in Equation (2) is to minimize the mutual information between video representation $h_v$ and the entire video $v$. By expanding this term, we have:

$$\text{MI}(h_v; v|s) = \mathbb{E}_{P(s)}\mathbb{E}_{P(v)}\Big[D_{\text{KL}}\big(\text{g}_\phi(h_v|s, v)||P(h_v|s)\big)\Big], \tag{3}$$

where $P(s)$ and $P(v)$ represent the state and video distribution, respectively. $P(h_v|s)$ is a marginal distribution $P(h_v|s) = \mathbb{E}_{P(v)}\text{g}_\phi(h_v|s, v)$. As estimating this marginal distribution could be intractable in practice, previous work (Goyal et al., 2018; Eysenbach et al., 2021) commonly approximates it with some prior $g(h_v|s)$. As we can see from Equation (3), the goal of this objective is to minimize the distance between the video representation produced by the video encoder

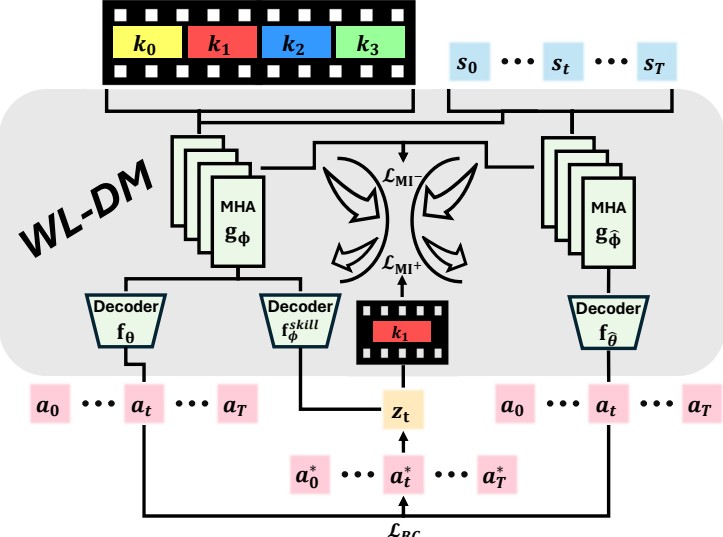

Figure 1: Overall Framework of WL-DM. We introduce an information bottleneck-based objective to achieve implicit video segmentation and skill discovery. Blocks with different colors represent different tasks. MHA stands for Multi-head Attention.

$h_v = \mathrm{g}_\phi(h_v|s, v)$ and some prior $g(h_v|s)$ that does not consider video $v$ at all. As shown later in the experiment, such a target for distance minimization is undesirable as it induces too much loss of video information. To solve this problem, we need to find a better alternative for $g(h_v|s)$ such that the loss of video information can be controlled at a proper level.

Recall the intuition in Section 4.1, we want the representation to be only related to the video segment of the current task $\mathrm{v}_{\mathrm{cur}}$. Although we cannot access the precise $\mathrm{v}_{\mathrm{cur}}$, we do have access to the video $v = (f_0, \cdots, f_T)$, which allows us to approximate $\mathrm{v}_{\mathrm{cur}}$ using a future video segment $\tilde{\mathrm{v}}_{\mathrm{cur}} = (f_t, f_{t+1}, \cdots f_{t+L})$ for state $s_t$, where $L$ is a randomly sampled window size. Therefore, we can use $\tilde{\mathrm{v}}_{\mathrm{cur}}$ as the input and get a prior video encoder $\mathrm{g}_{\tilde{\phi}}^{\mathrm{prior}}(h_v|s, \tilde{\mathrm{v}}_{\mathrm{cur}})$, which leads to the minimization of the following equation:

$$\mathbb{E}_{P(s)}\mathbb{E}_{P(v)}\Big[D_{\mathrm{KL}}\big(\mathrm{g}_\phi(h_v|s, v)|| \mathrm{g}_{\tilde{\phi}}^{\mathrm{prior}}(h_v|s, \tilde{\mathrm{v}}_{\mathrm{cur}})\big)\Big]. \tag{4}$$

With this prior encoder, we can get the final objective for mutual information minimization:

$$\mathcal{L}_{\mathrm{MI}^-} = \mathbb{E}_{s,v\sim D}\Big[D_{\mathrm{KL}}\big(\mathrm{g}_\phi(h_v|s, v)|| \mathrm{g}_{\tilde{\phi}}^{\mathrm{prior}}(h_v|s, \tilde{\mathrm{v}}_{\mathrm{cur}})\big)\Big]. \tag{5}$$

In addition to Equation (5), the prior encoder $\mathrm{g}_{\tilde{\phi}}^{\mathrm{prior}}$ is also trained via behavior cloning similar to Equation (1) with another action decoder $\mathrm{f}_{\tilde{\theta}}$ attached after it.

## 4.4 MAXIMIZING MUTUAL INFORMATION WITH SKILL APPROXIMATION

Another term in Equation (2) is to maximize the mutual information between the video representation $h_v$ and some skill approximation $z$. We follow Yuan et al. (2024) and use the short-term behavior $z = (a_t, a_{t+1}, \cdots, a_{t+M})$ as the representation of skills for state $s_t$, where $M$ is a randomly sampled window size. To enhance the level of abstraction of the skill representation, we further propose to first cluster all actions in the training dataset $\mathcal{D}$ and then use the cluster id $x_t$ of each action to improve the skill representation, such that $z = (x_t, x_{t+1}, \cdots, x_{t+M})$. As the mutual information is to measure the dependency between two variables, to maximize $\mathrm{MI}(h_v; z)$, we can simply maximize $\log P(z|h_v)$. As we have $z = (x_t, x_{t+1}, \cdots, x_{t+M})$, similar to Yuan et al. (2024), we can decompose the above maximization into each timestep and get the final objective for mutual information maximization:

$$\mathcal{L}_{\mathrm{MI}^+} = -\mathbb{E}_{s_t, x_t, v\sim\mathcal{D}}\Big[\log \mathrm{f}_\psi^{\mathrm{skill}}(x_t|s_t, \mathrm{g}_\phi(v))\Big], \tag{6}$$

where we introduce the skill decoder $f_\psi^{\text{skill}}$ to enhance the dependency between $h_t$ and $z$.

## 4.5 SUMMARY

Putting Equations (1), (5) and (6) together, we can now have the total loss for WL-DM:

$$\mathcal{L}_{\text{WL-DM}} = \mathcal{L}_{\text{BC}} + \alpha_1 \mathcal{L}_{\text{MI-}} + \alpha_2 \mathcal{L}_{\text{MI+}},$$

where we have two coefficients $\alpha_1$ and $\alpha_2$ to balance the scale of these three terms. The overall framework of our algorithm is illustrated in Figure 1. We use multiple self-attention layers as the encoder $g_\phi$ to process video tokens and state tokens and then use the action decoder $f_\theta$ to predict action labels $a_t^*$. The joint optimization of $\mathcal{L}_{\text{MI-}}$ and $\mathcal{L}_{\text{MI+}}$ ensures the video representation contains only information related to the current task. The pseudocode of our algorithm is summarized in Algorithm 1. It is worth noting that the skill decoder $f_\psi^{\text{skill}}$, the prior video encoder $g_{\tilde{\phi}}^{\text{prior}}$, and the prior action decoder $f_{\tilde{\theta}}$ will only be used during training, we will keep only the video encoder $g_\phi$ and the action decoder $f_\theta$ for execution.

---

**Algorithm 1** WL-DM

1: Initialize video encoder $g_\phi$, action decoder $f_\theta$ and skill decoder $f_\psi^{\text{skill}}$
2: Initialize prior video encoder $g_{\tilde{\phi}}^{\text{prior}}$ and prior action decoder $f_{\tilde{\theta}}$
3: Initialize training dataset $\mathcal{D}$
4: **for** $i = 1$ **to** $I$ **do**
5:     Sample data $(s_t, a_t^*, v)$ from $\mathcal{D}$
6:     Construct approximation of current video segment $v_{\text{cur}}$
7:     Construct approximation of current skill $z$
8:     Update $g_\phi$ and $f_\theta$ by Equation (1) with $(s_t, a_t^*, v)$
9:     Update $g_{\tilde{\phi}}^{\text{prior}}$ and $f_{\tilde{\theta}}$ by Equation (1) with $(s_t, a_t^*, v_{\text{cur}})$
10:    Update $g_\phi$ and $g_{\tilde{\phi}}^{\text{prior}}$ by Equation (5) with $(s_t, v, v_{\text{cur}})$
11:    Update $g_\phi$ and $f_\psi^{\text{skill}}$ by Equation (6) with $(s_t, z, v)$
12: **end for**

---

## 5 EXPERIMENTS

### 5.1 EXPERIMENT SETUP

To validate our method, we conduct empirical evaluations on two different robotic environments, Franka Kitchen (Gupta et al., 2020) and Meta World (Yu et al., 2020). The visualization of these two environments is presented in Figure 2.

In Franka Kitchen (**FK**), we control a Franka Panda robot in the kitchen environment to perform seven possible tasks: microwave (**M**), kettle (**K**), bottom burner (**B**), top burner (**T**), light switch (**L**), slide cabinet (**S**) and hinge cabinet (**H**). The dataset from the original paper (Gupta et al., 2020) contains 566 trajectories corresponding to 24 different task combinations. To enable video-conditioned policy training, we train expert policy using the

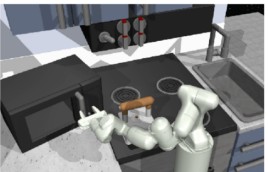 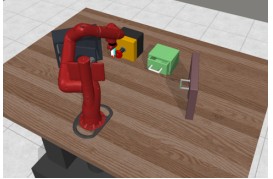

(a) Frank Kitchen        (b) Meta World

Figure 2: Visualization of Experiment Environments

original dataset to collect trajectories and corresponding video demonstrations. To evaluate the one-shot imitation learning ability, we split the dataset into a training dataset and a test dataset, where the training dataset contains 17 different task combinations and the test dataset contains 7 different task combinations, and there is no overlap of task combinations between the training set and the test set. During testing, we sample 3 different video demonstrations for each task combination, run the evaluation 10 times, and report the average performance.

In Meta world (**MW**), we modify the original environment (Yu et al., 2020) to perform multiple tasks within a single episode. In this newly devised environment, we control a Sawyer robot to perform four possible tasks: close drawer (**D**), open door (**O**), push button (**B**) and open window (**W**). We use expert policy to collect the dataset for all 24 different tasks. It is worth noting that, as the dataset contains all possible task combinations, the task orders presented in the video demonstration bring additional difficulties for policy learning. To evaluate the one-shot imitation learning ability, we split the dataset into a training dataset and a test dataset, where the training dataset contains 17 different task combinations and the test dataset contains 7 different task combinations, and there is no overlap of task combinations between the training set and the test set. During testing, we sample 3 different video demonstrations for each task combination, run the evaluation 10 times, and report the average performance.

We include several challenging imitation learning algorithms as our baselines: C-bet (Cui et al., 2022), decision transformer (Chen et al., 2021b), and VIMA (Jiang et al., 2023). As C-bet and decision transformer were not proposed for video-conditioned policy learning, we modify them to additionally take videos as input and get baselines **V-BET** and **V-DT**. For VIMA, it was originally proposed for multimodal prompts. However, as we do not assume data of other modalities, we train VIMA on our video-only dataset and serve as our baseline **VIMA**. More details of experiments can be found in Appendix A.

## 5.2 EXPERIMENT: MAIN

Table 1: The performance of WL-DM and baselines on all FK and MW tasks.

| Env | Methods | Tasks | | | | | | | Avg |
|-----|---------|------|------|------|------|------|------|------|-----|
| | | MBTH | MBLS | MBTL | KBTH | MTLH | KBLS | KBTS | |
| FK | WL-DM | **2.30** | **2.57** | **2.37** | 2.10 | **1.83** | **1.97** | 3.17 | $\mathbf{2.33}_{\pm 0.78}$ |
| | V-BET | 1.37 | 2.47 | 0.83 | **2.27** | 1.73 | 1.80 | 3.10 | $1.94_{\pm 1.06}$ |
| | V-DT | 1.00 | 2.33 | 1.70 | 1.33 | 1.47 | 1.70 | 2.10 | $1.66_{\pm 0.79}$ |
| | VIMA | 0.77 | 0.50 | 0.50 | 1.27 | 0.20 | 1.67 | 1.70 | $0.94_{\pm 0.85}$ |
| | | ODWB | DOBW | DBWO | WBOD | BDOW | BDWO | BWDO | |
| MW | WL-DM | **3.33** | 2.00 | **2.00** | **2.00** | **2.67** | **2.00** | **4.00** | $\mathbf{2.57}_{\pm 0.90}$ |
| | V-BET | 1.87 | 2.00 | 0.73 | 1.33 | 0.33 | 0.00 | 1.97 | $1.18_{\pm 0.85}$ |
| | V-DT | 1.33 | **2.13** | 1.23 | 1.93 | 0.37 | 0.83 | 0.83 | $1.24_{\pm 0.86}$ |
| | VIMA | 1.80 | 1.00 | 0.37 | 0.37 | 1.17 | 0.57 | 0.83 | $0.87_{\pm 0.84}$ |

As shown in Table 1, our method achieves better average performance in both environments. In Franka Kitchen, our method achieves an improvement of approximately 20.1% compared to the best baseline (V-BET). In Meta World, the improvement is even more significant, with our method achieving a 100.1% improvement compared to the best baseline (V-DT). Although our method consistently outperforms all baselines, we note that there is a large gap in terms of the degree of improvement between the two environments. This is because, in Franka Kitchen, task combinations in the dataset are not diverse enough, and there is no variation in terms of the task orders (A, B vs B, A). Therefore, it can be considered to have a strong correlation within the task combinations. Therefore, even without considering the segmentation of tasks in video demonstrations, we can still utilize this correlation to achieve a policy that performs well during testing. However, in Meta World, our dataset includes all task combinations and considers different task orders, making task segmentation in videos even more critical, which explains the gap in improvement of our algorithm in the two environments. More specifically, out of a total of 14 test tasks, our method achieves the best performance in 12 of them. Such overall performance validates the effectiveness of our algorithm.

For the performance of baselines, we found that V-BET and V-DT perform at a similar level. This is because the main difference between V-BET and V-DT in our implementation is whether or not action is used as part of the trajectory. For the robotic environments we used, this action information can often be directly inferred from changes in the state of the robot. Therefore, the advantage of using action information is not significant. VIMA, on the other hand, does not perform well in

both environments. One potential reason is that VIMA was originally proposed for multimodal scenarios. Although its training process can be transferred to the pure video scenario, this direct transfer is clearly not effective. Moreover, in terms of one-shot video imitation, VIMA mainly considers object-level variations in a single task rather than variations in task combinations, so we believe its performance decline is acceptable.

## 5.3 EXPERIMENT: ABLATION

Table 2: The performance of WL-DM and ablation baselines on all MW tasks.

| Env | Methods | Tasks | | | | | | | Avg |
| --- | --- | --- | --- | --- | --- | --- | --- | --- | --- |
| | | ODWB | DOBW | DBWO | WBOD | BDOW | BDWO | BWDO | |
| MW | WL-DM | **3.33** | **2.00** | **2.00** | **2.00** | **2.67** | **2.00** | **4.00** | **2.57**$_{\pm 0.90}$ |
| | WL-DM w/o $\mathcal{L}_{\text{MI+}}$ | 1.00 | **2.00** | 1.83 | 1.67 | 1.67 | 1.67 | 2.00 | 1.69$_{\pm 0.46}$ |
| | WL-DM w/o $\mathcal{L}_{\text{MI-}}$ | 2.00 | **2.00** | 2.00 | **2.00** | 1.67 | **2.00** | 3.33 | 2.14$_{\pm 0.64}$ |
| | WL-DM w $\varnothing$ | 2.00 | **2.13** | 1.93 | **2.00** | 2.00 | 1.33 | 2.00 | 1.91$_{\pm 0.37}$ |

As our objective function contains two different mutual information terms, we conduct ablation studies in this section to verify the contribution of these two components to our method. We construct two ablation baselines, WL-DM w/o $\mathcal{L}_{\text{MI+}}$ and WL-DM w/o $\mathcal{L}_{\text{MI-}}$. The ablation baselines are identical to our algorithm in all aspects, except that WL-DM w/o $\mathcal{L}_{\text{MI+}}$ does not use Equation (6) and WL-DM w/o $\mathcal{L}_{\text{MI-}}$ does not use Equation (5). We validate these two ablation baselines in the Meta World environment and compare them with our method. As shown in Table 2, both ablation baselines achieve worse performance compared to WL-DM, thereby verifying that both Equation (5) and Equation (6) contribute to our algorithm. Notably, even with only Equation (5), the ablation baseline still achieves better performance than the baselines in Section 5.2, which further validates the effectiveness of Theorem 1 in practice.

As mentioned in Section 4.3, directly minimizing the mutual information using prior $g(h_v|s)$ can lead to excessive loss of video information in practice, thereby affecting the performance of the algorithm. To verify this point, we construct another ablation baseline WL-DM w $\varnothing$. This baseline is again identical to our method in all aspects, except that it does not use $g_{\tilde{\phi}}^{\text{prior}}(h_v|s, \tilde{v}_{\text{cur}})$ but instead uses prior $g(h_v|s)$. From Table 2, we can see that using $g(h_v|s)$ indeed leads to a decline in the performance, thus verifying the prior video encoder we constructed in Section 4.3. Additionally, it is worth noting that WL-DM w/o $\mathcal{L}_{\text{MI-}}$ demonstrates that when we do not minimize mutual information at all, that is, do not control the information provided by the video, the algorithm cannot achieve its best performance. In contrast, WL-DM w $\varnothing$ indicates that excessively reducing the information of the video also leads to a decline in the final performance. This phenomenon further demonstrates the importance of a proper approximation for $v_{\text{cur}}$ and validates our statements in Section 4.3.

## 6 CONCLUSION

In this paper, we investigate the problem of one-shot video imitation for video-conditioned policies. To enhance the compositional generalization ability of the learned policy, we propose an imitation learning framework, **W**atch-**L**ess-**D**o-**M**ore (**WL-DM**). Our method introduces an information bottleneck-based objective, which leads to implicit skill discovery for video-conditioned policies. The intuition behind this method is that by segmenting the video into different tasks, the policy learns diverse skills corresponding to these tasks. When faced with unseen videos, the policy can also decompose them into combinations of previously encountered tasks, thereby completing these tasks through the learned skills. To better explain our method, we build a theoretical connection between our method and this intuition using information theory. We also present a practical implementation of our algorithm and evaluate it on a variety of tasks across multiple environments. The experimental results indicate that our algorithm outperforms baselines in terms of the compositional generalization ability, which verifies the effectiveness of our algorithm.

# 7 LIMITATIONS AND FUTURE WORK

The limitation of this work is that our approximations of the current task $k_{cur}$ and the current skill $z$ remain naive, which are just the information from the immediate future. Although similar approximations have been used in many previous studies (Pertsch et al., 2021; Liu et al., 2021; Xie et al., 2023; Yuan et al., 2024), one issue with this approach is the need for video data to be aligned with trajectory data in timesteps (for a state $s_t$, we can access its corresponding video frame $f_t$), which can be costly to collect in many cases.

Our future work will primarily focus on extending our algorithm to real-world scenarios, such as real-world robots, thereby broadening the application scope of our algorithm. Additionally, we will consider extending our algorithm to multimodal scenarios, utilizing multimodal information to obtain better approximations of tasks and skills, thereby not only enhancing the performance of the algorithm but also expanding the range of instruction formats it can process.

ACKNOWLEDGMENTS

This work was supported by NSFC under Grant 62450001 and 62476008. The authors would like to thank the anonymous reviewers for their valuable comments and advice.

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

## A    IMPLEMENTATION DETAILS

We implement our algorithm and all baselines based on the codebase of C-bet (Cui et al., 2022). For WL-DM and V-BET, we consider only observations in the trajectory, and for V-DT and VIMA, we consider both observations and actions in the trajectory, which aligns with the implementation stated in the paper of C-bet (Cui et al., 2022), DT Chen et al. (2021b) and VIMA (Jiang et al., 2023). For WL-DM, V-BET, and V-DT, we use the same transformer model as stated in C-bet, which contains multiple self-attention layers to process video information and trajectory information at the same time. For VIMA, we use alternating cross-attention and self-attention layers as described in its paper (Jiang et al., 2023).

For all experiments, we set the learning rate to be $3 \times 10^{-4}$ and set the window size for the trajectory to be 20 (for V-DT and VIMA, it means 20 observation-action pairs). For WL-DM, the window size of future video segments is sampled from $[20, 40]$. As we use the codebase of C-bet, all methods use the same action decoder, where we set the number of bins for action discretization to 32, and the id of each cluster will also be used for the representation of skills for WL-DM. For the Franka Kitchen environment (Gupta et al., 2020), we use decoders with 3 layers, and 3 heads and set the hidden dimension to be 60 (for VIMA, it means in total 3 self-attention layers and 3 cross-attention layers). We train all methods for 10 epochs. For WL-DM, $\alpha_1$ is fixed to be $1 \times 10^{-2}$ and $\alpha_2$ is fixed to be $1 \times 10^{-1}$ during the training process. For the Meta World environment (Yu et al., 2020), we use decoders with 6 layers, and 6 heads and set the hidden dimension to be 120 (for VIMA, it means in total 6 self-attention layers and 6 cross-attention layers). We train all methods for 30 epochs. For WL-DM, $\alpha_1$ is set to be 0 in the beginning and fixed to be $1 \times 10^{-3}$ after 10 epochs, and $\alpha_2$ is fixed to be 10 during the training process.

## B    PROOF OF THEOREM 1

**Theorem 1.** *If we have* $\mathrm{MI}(h_v; \mathrm{v_{other}} | s, \mathrm{v_{cur}}) = 0$, *then* $D_{\mathrm{KL}}\big(\pi(a|s, v) || \pi(a|s, \mathrm{v_{cur}})\big) = 0$ *for all state-video pairs* $(s, v) \in \mathcal{S} \times \mathcal{V}$ *with non-zero probability* $P(s, v) > 0$.

*Proof.* By expanding the mutual information $\mathrm{MI}(\mathrm{v_{other}}; h_v, a | s, \mathrm{v_{cur}})$, we can have the following equality:

$$
\begin{aligned}
&\mathrm{MI}(\mathrm{v_{other}}; h_v, a | s, \mathrm{v_{cur}}) \\
=&\mathbb{E}_{P(s, \mathrm{v_{cur}})} \mathbb{E}_{P(\mathrm{v_{other}}, h_v, a | s, \mathrm{v_{cur}})} \left[ \log \frac{P(\mathrm{v_{other}}, h_v, a | s, \mathrm{v_{cur}})}{P(\mathrm{v_{other}} | s, \mathrm{v_{cur}}) P(h_v, a | s, \mathrm{v_{cur}})} \right] \\
=&\mathbb{E}_{P(s, \mathrm{v_{cur}})} \mathbb{E}_{P(\mathrm{v_{other}}, h_v, a | s, \mathrm{v_{cur}})} \left[ \log P(h_v, a | s, \mathrm{v_{cur}}, \mathrm{v_{other}}) - \log P(h_v, a | s, \mathrm{v_{cur}}) \right] \\
=&\mathbb{E}_{P(s, \mathrm{v_{cur}})} \mathbb{E}_{P(\mathrm{v_{other}}, h_v, a | s, \mathrm{v_{cur}})} \left[ \begin{matrix} \log P(h_v | s, \mathrm{v_{cur}}, \mathrm{v_{other}}) + P(a | h_v, s, \mathrm{v_{cur}}, \mathrm{v_{other}}) \\ - \log P(h_v | s, \mathrm{v_{cur}}) - \log P(a | h_v, s, \mathrm{v_{cur}}) \end{matrix} \right] \\
=&\mathbb{E}_{P(s, \mathrm{v_{cur}}, \mathrm{v_{other}})} \left[ D_{\mathrm{KL}}(P(h_v | s, \mathrm{v_{cur}}, \mathrm{v_{other}}) || P(h_v | s, \mathrm{v_{cur}})) \right] \\
&+ \mathbb{E}_{P(h_v, s, \mathrm{v_{cur}}, \mathrm{v_{other}})} \left[ D_{\mathrm{KL}}(P(a | h_v, s, \mathrm{v_{cur}}, \mathrm{v_{other}}) || P(a | h_v, s, \mathrm{v_{cur}})) \right] \\
=& \mathrm{MI}(h_v; \mathrm{v_{other}} | s, \mathrm{v_{cur}}) + \mathrm{MI}(a; \mathrm{v_{other}} | s, \mathrm{v_{cur}}, h_v).
\end{aligned}
$$

Similarly, we can also have:

$$
\mathrm{MI}(\mathrm{v_{other}}; h_v, a | s, \mathrm{v_{cur}}) = \mathrm{MI}(a; \mathrm{v_{other}} | s, \mathrm{v_{cur}}) + \mathrm{MI}(h_v; \mathrm{v_{other}} | s, \mathrm{v_{cur}}, a).
$$

Combining these two equality, we can have:

$$
\begin{aligned}
&\mathrm{MI}(h_v; \mathrm{v_{other}} | s, \mathrm{v_{cur}}) + \mathrm{MI}(a; \mathrm{v_{other}} | s, \mathrm{v_{cur}}, h_v) \\
=& \mathrm{MI}(a; \mathrm{v_{other}} | s, \mathrm{v_{cur}}) + \mathrm{MI}(h_v; \mathrm{v_{other}} | s, \mathrm{v_{cur}}, a).
\end{aligned}
$$

As $a$ and $\mathrm{v_{other}}$ become independent with each other when $h_v$ is given, we have $\mathrm{MI}(a; \mathrm{v_{other}} | s, \mathrm{v_{cur}}, h_v) = 0$. As we also have $\mathrm{MI}(h_v; \mathrm{v_{other}} | s, \mathrm{v_{cur}}, a) \geq 0$, we can have the

following inequality, which basically gives us the conditional version of data processing inequality (Cover, 1999):

$$\mathrm{MI}(h_v; \mathrm{v}_{\mathrm{other}} \,|\, s, \mathrm{v}_{\mathrm{cur}}) \geq \mathrm{MI}(a; \mathrm{v}_{\mathrm{other}} \,|\, s, \mathrm{v}_{\mathrm{cur}}).$$

Since $\mathrm{MI}(a; \mathrm{v}_{\mathrm{other}} \,|\, s, \mathrm{v}_{\mathrm{cur}}) \geq 0$, if we can also have $\mathrm{MI}(h_v; \mathrm{v}_{\mathrm{other}} \,|\, s, \mathrm{v}_{\mathrm{cur}}) = 0$, then we can conclude that:

$$\mathrm{MI}(a; \mathrm{v}_{\mathrm{other}} \,|\, s, \mathrm{v}_{\mathrm{cur}}) = 0.$$

By expanding this mutual information term, we have:

$$\begin{aligned}
&\mathrm{MI}(a; \mathrm{v}_{\mathrm{other}} \,|\, s, \mathrm{v}_{\mathrm{cur}}) \\
=& \mathbb{E}_{P(s, \mathrm{v}_{\mathrm{cur}}, \mathrm{v}_{\mathrm{other}})} \Big[ D_{\mathrm{KL}}(\pi(a|s, \mathrm{v}_{\mathrm{cur}}, \mathrm{v}_{\mathrm{other}}) || \pi(a|s, \mathrm{v}_{\mathrm{cur}})) \Big] \\
=& \mathbb{E}_{P(s, v)} \Big[ D_{\mathrm{KL}}(\pi(a|s, v) || \pi(a|s, \mathrm{v}_{\mathrm{cur}})) \Big] \\
=& 0.
\end{aligned}$$

Since the KL divergence is non-negative, for the above expectation to be zero, there must be for all state-video pairs $(s, v) \in \mathcal{S} \times \mathcal{V}$ with non-zero probability $P(s, v) > 0$, we have the KL divergence to be zero, $D_{\mathrm{KL}}(\pi(a|s, v) || \pi(a|s, \mathrm{v}_{\mathrm{cur}})) = 0$, and conclude our proof. □

## C   VISUALIZATION

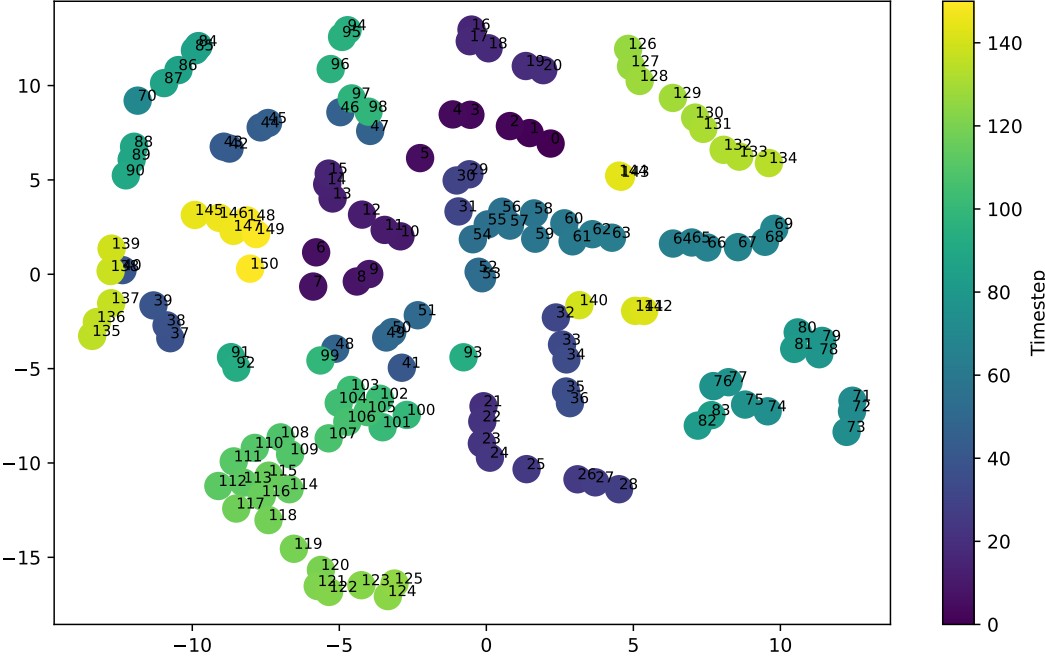

Figure 3: Visualization of $h_v$ over timesteps.

In this section, we present the visualization result of our method. We visualize how $h_v$ of WL-DL changes over timesteps. As shown in Figure 3, we can observe that $h_v$ of WL-DM tends to converge at adjacent timesteps. It is worth noting that since we use a GPT-like transformer architecture as the encoder, the information of video tokens and obs tokens are mixed together in $h_v$. Furthermore, we do not introduce any task-level information (such as task-level video segmentation annotations), so the clustering results of $h_v$ do not fully correspond to the task.

# D ADDITIONAL EXPERIMENTS

## D.1 ABLATION: TYPES OF TASK COMBINATIONS

We further include an experiment about the effect of the number of task combinations in the training set in the Meta World environment. In Section 5.2, we included 17 task combinations (7/3 split) in the training set, and here we further consider cases where we have 15 task combinations (6/4 split) and 20 task combinations in the training set. As shown Table 3, experimental results, WL-DM still outperforms other baselines, further demonstrating the effectiveness of our method.

|  | WL-DM | V-BET | V-DT | VIMA |
|---|---|---|---|---|
| 6/4 | $\mathbf{1.93}_{\pm 0.26}$ | $1.34_{\pm 0.78}$ | $0.86_{\pm 0.84}$ | $0.43_{\pm 0.78}$ |
| 7/3 (main exp) | $\mathbf{2.57}_{\pm 0.90}$ | $1.18_{\pm 0.85}$ | $1.24_{\pm 0.86}$ | $0.87_{\pm 0.84}$ |
| 8/2 | $\mathbf{1.88}_{\pm 0.36}$ | $0.63_{\pm 0.86}$ | $0.94_{\pm 0.88}$ | $0.81_{\pm 0.61}$ |

Table 3: The performance of all methods with different number of task combinations in the training set on MW tasks.

## D.2 ABLATION: NUMBER OF VIDEOS FOR EACH TASK COMBINATION

We also include an experiment about the number of videos corresponding to each task combination in the Meta World environment. In Section 5.2 we considered 20 different videos for each task combination, and here we further consider cases with 40 different videos for each task combination. As shown Table 4, experimental results, WL-DM still outperforms other baselines, which again demonstrates the effectiveness of WL-DM.

|  | WL-DM | V-BET | V-DT | VIMA |
|---|---|---|---|---|
| 20 (main exp) | $\mathbf{2.57}_{\pm 0.90}$ | $1.18_{\pm 0.85}$ | $1.24_{\pm 0.86}$ | $0.87_{\pm 0.84}$ |
| 40 | $\mathbf{2.21}_{\pm 0.81}$ | $1.71_{\pm 0.63}$ | $1.33_{\pm 0.90}$ | $1.09_{\pm 0.63}$ |

Table 4: The performance of all methods with different number of videos for each task combination on MW tasks.

## D.3 MORE BASELINE: VIP

| Env | Methods | Tasks | | | | | | | Avg |
|---|---|---|---|---|---|---|---|---|---|
|  |  | ODWB | DOBW | DBWO | WBOD | BDOW | BDWO | BWDO |  |
| MW | WL-DM | **3.33** | 2.00 | **2.00** | **2.00** | **2.67** | 2.00 | 4.00 | $\mathbf{2.57}_{\pm 0.90}$ |
|  | V-BET | 1.87 | 2.00 | 0.73 | 1.33 | 0.33 | 0.00 | 1.97 | $1.18_{\pm 0.85}$ |
|  | V-DT | 1.33 | 2.13 | 1.23 | 1.93 | 0.37 | 0.83 | 0.83 | $1.24_{\pm 0.86}$ |
|  | VIMA | 1.80 | 1.00 | 0.37 | 0.37 | 1.17 | 0.57 | 0.83 | $0.87_{\pm 0.84}$ |
|  | ViP | 2.80 | **2.20** | **2.00** | 1.87 | 1.63 | 1.10 | 1.80 | $1.91_{\pm 0.88}$ |

Table 5: The performance of all methods including ViP on all MW tasks.

We have added a new video-conditioned baseline: ViP (Chane-Sane et al., 2023). Although the purpose of ViP is to learn a video-conditioned policy, it has a different setting from our method. Thus, we have made the following modifications to adapt it to our setting:

- Since we do not consider human videos as input, we have removed the part that uses human videos for pre-training.

- Since we do not assume access to video labels, we have changed its supervised contrastive learning part to unsupervised contrastive learning on robot videos.

We used the same codebase as WL-DM to implement ViP with minimal modifications, and conducted experiments in the MetaWorld environment. The experimental results are shown in Table 5. ViP outperformed other baselines in this environment, demonstrating its effectiveness as video-conditioned policy. However, its performance still lags behind WL-DM, which further demonstrates the effectiveness of WL-DM.

## D.4 MORE DATASET

To further evaluate our method, we construct script in a similar way of Lee et al. (2024) to convert state-base observations of the original dataset of the Franka Kitchen environment into image-based observation, and train all methods on this dataset. For WL-DM, we use a linear schedule for $\alpha_1$, where coef_start is set to be 0 and coef_end is set to be $1 \times 10^{-4}$, and for $\alpha_2$, we fix it to be $1 \times 10^{-1}$ during the training process. As shown in Table 6, WL-DM still outperforms other methods, which further demonstrates the effectiveness of our method.

| Env | Methods | Tasks | | | | | | | Avg |
|-----|---------|-------|------|------|------|------|------|------|-----|
| | | BTLS | BTSH | MBTS | MBTH | MLSH | MBTL | MKBH | |
| FK(new) | WL-DM | **2.43** | 1.63 | **2.63** | **2.57** | **2.27** | **2.27** | **2.5** | $\mathbf{2.33}_{\pm 0.74}$ |
| | V-BET | 1.23 | 1.43 | 2.33 | 1.53 | 2.10 | 1.70 | 2.17 | $1.79_{\pm 0.80}$ |
| | V-DT | 1.63 | **2.20** | 1.40 | 1.80 | 1.47 | 1.73 | 2.20 | $1.78_{\pm 0.83}$ |
| | VIMA | 0.87 | 1.27 | 2.20 | 1.80 | 1.80 | 1.43 | 2.23 | $1.66_{\pm 0.80}$ |

Table 6: The performance of all methods on new FK dataset.

## D.5 MORE BASE ALGORITHM

As WL-DM can be seen as a method using information bottleneck-based loss on top of V-BET. To further validate our approach, we applied the information bottleneck-based loss of WL-DM to both V-DT and VIMA and conducted experiments in the Meta World environment. As shown in Table 7, WL-DM+V-DT and WL-DM+VIMA both outperform its base algorithm, which further validates the effectiveness of the proposed information bottleneck-based loss.

| | WL-DM | V-BET | WL-DM+V-DT | V-DT | WL-DM+VIMA | VIMA |
|---|-------|-------|------------|------|------------|------|
| MW | $\mathbf{2.57}_{\pm 0.90}$ | $1.18_{\pm 0.85}$ | $1.72_{\pm 0.68}$ | $1.24_{\pm 0.86}$ | $1.84_{\pm 0.84}$ | $0.87_{\pm 0.84}$ |

Table 7: The performance of all methods on MW tasks, we applied the information bottleneck-based loss on different base algorithms and compare their performance.

## D.6 COEFFICIENT SELECTION

| $\alpha_1$ \ $\alpha_2$ | 0.1 | 1 | 10 |
|---|-----|-----|-----|
| 0.1 | $0.95_{\pm 0.77}$ | $2.09_{\pm 0.69}$ | $2.16_{\pm 0.71}$ |
| 0.01 | $0.82_{\pm 0.86}$ | $1.83_{\pm 0.53}$ | $2.05_{\pm 0.62}$ |
| 0.001 | $1.03_{\pm 0.80}$ | $1.79_{\pm 0.41}$ | $\mathbf{2.57}_{\pm 0.90}$ |

Table 8: The performance of WL-DM with different coefficients on MW tasks.

Coefficients for mutual information loss should be adjusted according to the environment. Specifically, we conducted a grid search to select optimal coefficients. Taking experiments in Meta World environment as an example, the performance of different coefficients is shown in Table 8.

