# OpenReview forum: "Watch Less, Do More: Implicit Skill Discovery for Video-Conditioned Policy"
_ICLR.cc/2025/Conference — ICLR 2025 Poster_

### Official Review · Reviewer_wcsD · 2024-10-28

**Soundness:** 3
**Presentation:** 2
**Contribution:** 2
**Rating:** 8
**Confidence:** 4

**Summary:**

The paper introduces Watch-Less-Do-More (WL-DM), an imitation learning framework designed for video-conditioned policy learning that aims to enhance compositional generalization. WL-DM uses an information bottleneck method to identify relevant skills from videos, allowing the policy to handle complex, unseen video task combinations by focusing only on current tasks. Experimental results across two robotic environments (Franka Kitchen and Meta World) show WL-DM surpassing baseline models in generalization ability. The authors also highlight the potential for broader applications, though they note limitations regarding video data alignment with task segmentation.

**Strengths:**

- The framework is sound, well-structured, and novel.
- The overall content is easy to understand and well-written.

**Weaknesses:**

- It would be beneficial to clearly explain the difference between the proposed method and various existing approaches that use mutual information for skill extraction [1, 2, 3].
- There are questions regarding the appropriateness of the chosen baselines. The authors used VIMA as the SOTA baseline, but VIMA showed very low performance. They suggest that this may be due to VIMA's reliance on multi-modal data, which could degrade performance when using only pure video data. However, a comparison with other one-shot imitation methods that use only video data would provide a more relevant evaluation.

[1] Jiang, Yiding, et al. "Learning options via compression." *Advances in Neural Information Processing Systems* 35 (2022): 21184-21199.

[2] Ju, Zhaoxun, et al. "Rethinking Mutual Information for Language Conditioned Skill Discovery on Imitation Learning." *Proceedings of the International Conference on Automated Planning and Scheduling*. Vol. 34. 2024.

[3] Yu, Xuehui, et al. "Skill-aware Mutual Information Optimisation for Generalisation in Reinforcement Learning." *arXiv preprint arXiv:2406.04815* (2024).

**Questions:**

- For mutual information loss, the importance might vary depending on the diversity and length of skills in the environment. Should it be adjusted according to each environment? If so, how should it be approached?
- Could you show how the skills were separated in the experiments conducted? Do similar z-values actually appear consecutively, allowing the skills to be segmented?
- Additionally, providing a visual representation of whether the skill space is discretely separated into a minimal set of skills when using this type of loss would aid the reader’s understanding.

---

> ### Author Response · Authors · 2024-11-23
> **Official Comment to Reviewer wcsD (1/2)**
>
> ## Weakness 1
> It would be beneficial to clearly explain the difference between the proposed method and various existing approaches that use mutual information for skill extraction [1, 2, 3].
> ## Answer 1
> Thank you for pointing out these related works. The main difference between our work and these papers lies in the problem we are addressing. In this paper, we focus on implicitly discovering skills from videos, such that the trained video-conditioned policy can achieve zero-shot compositional generalization.
>
> In [1], the authors consider learning skills from pre-collected experiences to accelerate the learning of new tasks. In contrast to [1], our goal is to learn a policy with zero-shot compositional generalization capabilities without the need for training on new tasks.
>
> In [2], the authors address skill discovery under the problem of language-conditioned policy learning. Compared to [2], we consider video-conditioned policy learning and the implicit skill discovery from videos.
>
> In [3], the authors tackle the Meta-RL problem by learning context embeddings to generalize to changes in environmental features. In our paper, we focus on implicitly discovering skills from videos to generalize to unseen task combinations.
>
> ## Weakness 2
> There are questions regarding the appropriateness of the chosen baselines. The authors used VIMA as the SOTA baseline, but VIMA showed very low performance. They suggest that this may be due to VIMA's reliance on multi-modal data, which could degrade performance when using only pure video data. However, a comparison with other one-shot imitation methods that use only video data would provide a more relevant evaluation.
> ## Answer 2
> Thank you for your suggestion, a comparison with other one-shot imitation methods would indeed strengthen our paper. We have added a new one-shot imitation baseline: ViP[4], which takes only the task video as the one-shot demonstration. Although the purpose of ViP is also to learn a video-conditioned policy, it has a different setting from our method. Thus, we have made the following modifications to adapt it to our setting:
> - Since we do not consider human videos as input, we have removed the part that uses human videos for pre-training.
> - Since we do not assume access to video labels, we have changed its supervised contrastive learning part to unsupervised contrastive learning on robot videos.
>
> We used the same codebase as WL-DM to implement ViP with minimal modifications and conducted experiments in the MetaWorld environment. The experimental results are as follows. ViP outperformed other baselines in this environment, demonstrating its effectiveness as one-shot imitation baseline. However, its performance still lags behind WL-DM, which further demonstrates the effectiveness of our method.
>
> |   Env  /  Method  |        WL-DM         |        V-BET      |        V-DT       |        VIMA       |        ViP        |
> |:-----------------:|:--------------------:|:-----------------:|:-----------------:|:-----------------:|:-----------------:|
> |      MetaWorld    |$\textbf{2.57}\pm0.90$|    $1.18\pm0.85$  |   $1.24\pm0.86$   |   $0.87\pm0.84$   |   $1.91\pm0.88$   |
>
> [1] Jiang, Yiding, et al. "Learning options via compression." Advances in Neural Information Processing Systems 35 (2022): 21184-21199.
>
> [2] Ju, Zhaoxun, et al. "Rethinking Mutual Information for Language Conditioned Skill Discovery on Imitation Learning." Proceedings of the International Conference on Automated Planning and Scheduling. Vol. 34. 2024.
>
> [3] Yu, Xuehui, et al. "Skill-aware Mutual Information Optimisation for Generalisation in Reinforcement Learning." arXiv preprint arXiv:2406.04815 (2024).
>
> [4] Chane-Sane, Elliot, et al. "Learning video-conditioned policies for unseen manipulation tasks." 2023 IEEE International Conference on Robotics and Automation (ICRA). IEEE, 2023.

---

> ### Author Response · Authors · 2024-11-23
> **Official Comment to Reviewer wcsD (2/2)**
>
> ## Question 1
> For mutual information loss, the importance might vary depending on the diversity and length of skills in the environment. Should it be adjusted according to each environment? If so, how should it be approached?
> ## Answer 3
> Yes, coefficients for mutual information loss should be adjusted according to the environment. Specifically, we conducted a grid search to select optimal coefficients. Taking experiments in MetaWorld environment as an example, the performance of different coefficients is shown in the following table.
>
> |$\alpha_1$/$\alpha_2$|        0.1           |        1          |        10            |
> |:-------------------:|:--------------------:|:-----------------:|:--------------------:|
> |      0.1            |     $0.95\pm0.77$    |   $2.09\pm0.69$   |    $2.16\pm0.71$     |
> |      0.01           |     $0.82\pm0.86$    |   $1.83\pm0.53$   |    $2.05\pm0.62$     |
> |      0.001          |     $1.03\pm0.80$    |   $1.79\pm0.41$   |$\textbf{2.57}\pm0.90$|
>
> ## Question 2
> Could you show how the skills were separated in the experiments conducted? Do similar z-values actually appear consecutively, allowing the skills to be segmented?
> ## Answer 4
> As we mentioned in Lines 329-331, we did not really use $z$ in our method, we decompose it into each timestep and predict $x_t$ instead. So we don't really have a representation of skill in our method to visualize.
>
> ## Question 3
> Additionally, providing a visual representation of whether the skill space is discretely separated into a minimal set of skills when using this type of loss would aid the reader’s understanding.
> ## Answer 5
> Thanks for your suggestion. As for visualizations, we have added the visualization result of WL-DM in our revised paper, Appendix C. We visualized how $h_v$ of WL-DL changes over timesteps. From the figure, we can observe that $h_v$ of WL-DM tends to converge at adjacent timesteps. It is worth noting that since we used a GPT-like transformer architecture as the encoder, the information of video tokens and obs tokens are mixed together in $h_v$. Furthermore, we did not introduce any task-level information (such as task-level video segmentation annotations), so the clustering results of $h_v$ do not fully correspond to the task.

---

> > ### Comment · Reviewer_wcsD · 2024-11-24
> > **Thank you for your detailed response**
> >
> > Thank you for your detailed response. The inclusion of additional baselines and visualizations has greatly enhanced the quality of the paper. Accordingly, I have updated the score to reflect the points addressed in the rebuttal.

---

> > > ### Author Response · Authors · 2024-11-24
> > > **Thanks for your response**
> > >
> > > Thank you for taking the time to review our paper and provide suggestions. Your feedback has greatly helped us improve our work.

---

### Official Review · Reviewer_88xK · 2024-10-31

**Soundness:** 3
**Presentation:** 3
**Contribution:** 2
**Rating:** 6
**Confidence:** 4

**Summary:**

The paper proposes an information bottleneck-based imitation learning framework, WL-DM (Watch-Less-Do-More), for implicit skill discovery and video-conditioned policy learning. This strategy can generalize to unseen video task combinations, demonstrating strong compositional generalization ability. The idea of implicit skill decomposition is innovative, especially in achieving task segmentation within videos without requiring explicit video segmentation annotations.

**Strengths:**

This paper proposes that the WL-DM framework innovatively employs an information bottleneck approach for implicit skill discovery, allowing effective task segmentation without explicit annotations. The method demonstrates strong compositional generalization, successfully adapting to unseen video task combinations.

**Weaknesses:**

1. The compositional generalization seems limited to combining different tasks, while in practice, different task combination sequences will not have different effects on similar outcomes. Could the authors clarify the practical significance of this generalization with examples of real-world scenarios where it proves valuable or challenging?

2. The paper claims that video-conditioned policies can achieve combinatorial generalization when tasks can be performed independently. Could you explicitly compare WL-DM to single-task learning methods and clarify which mechanisms in WL-DM enable combinatorial generalization beyond what single-task approaches offer?

3. The paper lacks an experimental comparison with skill-based imitation learning methods (e.g., Xu et al., 2023; Wang et al., 2023; Shin et al., 2023; 2024) and single-task video demonstration methods (e.g., Chane-Sane et al., 2023). Could the authors analyze specific performance metrics, such as generalization ability or data efficiency, to provide a more detailed comparison with these methods?

4. The experimental results are relatively limited, such as lacking empirical support for the advantages of implicit segmentation, as well as necessary visualizations or other forms of demonstrations to validate the effects of implicit segmentation.

**Questions:**

1. what is the practical significance of compositional generalization?

2. Can the author analyze the performance of the above paper methods in the experiment?

3. Can the author provide more experimental results to verify the effectiveness of the model?

---

> ### Author Response · Authors · 2024-11-23
> **Official Comment to Reviewer 88xK (1/3)**
>
> ## Weakness 1
> The compositional generalization seems limited to combining different tasks, while in practice, different task combination sequences will not have different effects on similar outcomes. Could the authors clarify the practical significance of this generalization with examples of real-world scenarios where it proves valuable or challenging?
> ## Answer 1
>
>     The compositional generalization seems limited to combining different tasks, while in practice, different task combination sequences will not have different effects on similar outcomes.
>
> It is true that considering different execution sequences of the same set of tasks (e.g., ABCD vs ABDC) may lead to similar outcomes. However, the scenario of compositional generalization extends beyond this. Different task combinations may involve different sets of tasks, which in turn can produce different outcomes (e.g., ABCD vs ABEF).
>
> For instance, in the Franka kitchen environment, the policy is required to complete the tasks based on the task combinations demonstrated in the videos. There are seven possible tasks, but each video demonstration only includes four of them. Therefore, the policy needs to handle different task combinations such as (microwave, kettle, light switch, slide cabinet) and (microwave, kettle, bottom burner, top burner), and the effects of completing these two task combinations in the environment are clearly different.
>
>     Could the authors clarify the practical significance of this generalization with examples of real-world scenarios where it proves valuable or challenging?
>
>
> Thank you for your suggestion. There are indeed many real-world scenarios where compositional generalization is applicable. For example, in the scenario of industrial assembly lines, as productions are often highly modularized, we do not need to deal with an infinite number of tasks, but rather combinations of a finite number of tasks. In such a scenario, an algorithm that achieves compositional generalization can greatly reduce the cost of collecting training data, as we no longer need a training set that covers all task combinations, but only one that covers all individual tasks.
>
> In addition to the practical significance of compositional generalization, it is also valuable as a research topic. This problem has been widely studied in multiple fields, such as Visual Generation[1], Vision-Language Model[2], and Semantic Parsing[3]. Considering both its practical significance and research value, we believe that our study on compositional generalization is meaningful to some extent.
>
> We appreciate your suggestion and will include corresponding text in the revised paper to clarify the significance of compositional generalization.
>
> [1] Liu, Nan, et al. "Compositional visual generation with composable diffusion models." European Conference on Computer Vision. Cham: Springer Nature Switzerland, 2022.
>
> [2] Li, Chuanhao, et al. "In-context compositional generalization for large vision-language models." Proceedings of the 2024 Conference on Empirical Methods in Natural Language Processing. 2024.
>
> [3] Oren, Inbar, et al. "Improving Compositional Generalization in Semantic Parsing." Findings of the Association for Computational Linguistics: EMNLP 2020. 2020.

---

> ### Author Response · Authors · 2024-11-23
> **Official Comment to Reviewer 88xK (2/3)**
>
> ## Weakness 2
> The paper claims that video-conditioned policies can achieve combinatorial generalization when tasks can be performed independently. Could you explicitly compare WL-DM to single-task learning methods and clarify which mechanisms in WL-DM enable combinatorial generalization beyond what single-task approaches offer?
> ##  Answer 2
> For the single-task learning method, as we mentioned in Lines 217-219 of our paper, obtaining task-level video segmentation annotations is often difficult in many cases. Therefore, in our work, we assume that such video segmentation annotations do not exist and thus can not construct the training set for single-task learning methods. Additionally, when dealing with videos that contain multiple tasks, a single-task policy trained on single-task videos would still require an additional mechanism or model to determine which tasks are included in the video and when to switch between tasks (like the method used in Vid2Robot[1]). Therefore, we did not include such a baseline for comparison.
>
> For WL-DM, as stated in Lines 219-222 of our paper, we introduce an information-bottleneck based loss function, ensuring that the video representation only contains information related to the current task. This allows the policy to focus on the current task when making decisions. Since each single task is covered in the training set, WL-DM is able to decompose unseen task combinations into individual tasks that have been encountered before, thereby achieving compositional generalization.
>
> ## Weakness 3
> The paper lacks an experimental comparison with skill-based imitation learning methods (e.g., Xu et al., 2023; Wang et al., 2023; Shin et al., 2023; 2024) and single-task video demonstration methods (e.g., Chane-Sane et al., 2023). Could the authors analyze specific performance metrics, such as generalization ability or data efficiency, to provide a more detailed comparison with these methods?
> ## Answer 3
> Thank you for your suggestion. As we mentioned in Lines 51-53, these skill-based imitation learning methods[2,3,4,5] require explicit video segmentation annotations, or videos of another embodiment to train a skill-based policy. Therefore, we did not include them as our baseline.
>
> For single-task video demonstration methods, we have added a new video-conditioned baseline: ViP[6]. Although the purpose of ViP is to learn a video-conditioned policy, it has a different setting from our method. Thus, we have made the following modifications to adapt it to our setting:
> - Since we do not consider human videos as input, we removed the part that uses human videos for pre-training.
> - Since we do not assume access to video labels, we changed its supervised contrastive learning component to unsupervised contrastive learning on robot videos.
>
> We used the same codebase as WL-DM to implement ViP with minimal modifications and conducted experiments in the MetaWorld environment to evaluate the generalization ability of different algorithms. The experimental results are as follows. ViP outperformed other baselines in this environment, demonstrating its effectiveness as a video-conditioned policy. However, its performance still lags behind WL-DM, which further demonstrates the effectiveness of our method.
>
> |   Env  /  Method  |        WL-DM         |        V-BET      |        V-DT       |        VIMA       |        ViP        |
> |:-----------------:|:--------------------:|:-----------------:|:-----------------:|:-----------------:|:-----------------:|
> |      MetaWorld    |$\textbf{2.57}\pm0.90$|    $1.18\pm0.85$  |   $1.24\pm0.86$   |   $0.87\pm0.84$   |   $1.91\pm0.88$   |
>
> [1] Jain, Vidhi, et al. "Vid2robot: End-to-end video-conditioned policy learning with cross-attention transformers." arXiv preprint arXiv:2403.12943 (2024).
>
> [2] Xu, Mengda, et al. "Xskill: Cross embodiment skill discovery." Conference on Robot Learning. PMLR, 2023.
>
> [3] Wang, Chen, et al. "MimicPlay: Long-Horizon Imitation Learning by Watching Human Play." Conference on Robot Learning. PMLR, 2023.
>
> [4] Shin, Sangwoo, et al. "One-shot imitation in a non-stationary environment via multi-modal skill." International Conference on Machine Learning. PMLR, 2023.
>
> [5] Shin, Sangwoo, et al. "SemTra: A Semantic Skill Translator for Cross-Domain Zero-Shot Policy Adaptation." Proceedings of the AAAI Conference on Artificial Intelligence. Vol. 38. No. 13. 2024.
>
> [6] Chane-Sane, Elliot, et al. "Learning video-conditioned policies for unseen manipulation tasks." 2023 IEEE International Conference on Robotics and Automation (ICRA). IEEE, 2023.

---

> ### Author Response · Authors · 2024-11-23
> **Official Comment to Reviewer 88xK (3/3)**
>
> ## Weakness 4
> The experimental results are relatively limited, such as lacking empirical support for the advantages of implicit segmentation, as well as necessary visualizations or other forms of demonstrations to validate the effects of implicit segmentation.
> ## Answer 4
> Thank you for your insightful comments. Regarding the advantages of implicit segmentation, it is important to note that its strength lies in the ability to train policy that achieves compositional generalization without the need for annotated video segmentation. Therefore, its advantage is not related to the empirical evaluation of performance but rather to the reduced requirements for the training dataset.
>
> As for visualizations, we have added the visualization result of WL-DM in our revised paper, Appendix C. We visualized how $h_v$ of WL-DL changes over timesteps. From the figure, we can observe that $h_v$ of WL-DM tends to converge at adjacent timesteps. It is worth noting that since we used a GPT-like transformer architecture as the encoder, the information of video tokens and obs tokens are mixed together in $h_v$. Furthermore, we did not introduce any task-level information (such as task-level video segmentation annotations), so the clustering results of $h_v$ do not fully correspond to the task.
>
> WL-DM can be seen as using an information bottleneck-based loss function on top of V-BET. To further validate our approach, we applied the information bottleneck-based loss function of WL-DM to both V-DT and VIMA and conducted experiments in the MetaWorld environment.
>
> |   Env  /  Method  |       WL-DM          |      V-BET        |    WL-DM+V-DT     |       V-DT        |    WL-DM+VIMA     |        VIMA       |
> |:-----------------:|:--------------------:|:-----------------:|:-----------------:|:-----------------:|:-----------------:|:-----------------:|
> |      MetaWorld    |$\textbf{2.57}\pm0.90$|   $1.18\pm0.85$   |   $1.72\pm0.68$   |   $1.24\pm0.86$   |    $1.84\pm0.84$  |    $0.87\pm0.84$  |
>
> As shown in the above experimental results, WL-DM+V-DT and WL-DM+VIMA both outperform their base algorithm, which further validates the effectiveness of our method.
>
> ## Question 1
> what is the practical significance of compositional generalization?
> ## Answer 5
> See Answer 1
>
> ## Question 2
> Can the author analyze the performance of the above paper methods in the experiment?
> ## Answer 6
> see Answer 3
>
> ## Question 3
> Can the author provide more experimental results to verify the effectiveness of the model?
> ## Answer 7
> See Answer 4

---

> ### Comment · Reviewer_88xK · 2024-11-25
>
> Thanks for your responses to address my concerns. I decided to maintain my score considering the quality of this paper, other reviews, and the authors' responses. I tend to accept this paper.

---

> > ### Author Response · Authors · 2024-11-25
> > **Thanks for your response**
> >
> > Thank you for taking the time to review our paper and provide suggestions. Your feedback has greatly helped us improve our work.

---

### Official Review · Reviewer_DKw6 · 2024-11-04

**Soundness:** 3
**Presentation:** 3
**Contribution:** 2
**Rating:** 6
**Confidence:** 3

**Summary:**

This paper proposes a new method to train video-conditioned policies, which employ an information bottleneck-based objective to learn a video encoder for implicit skill discovery. The proposed method is novel, and the experiments on both Frank Kitchen and MetaWorld demonstrate its performance outperforms baselines.

**Strengths:**

- Learning an executable policy from videos is a good topic, since video is a general interface across different domains and there are widely available video data. This paper further proposes an information bottleneck-based imitation learning framework for implicit skill discovery and video-conditioned policy learning, which is novel and sound.
- Good writing and clear motivation. This paper derives its final objective from both skill discovery and information bottleneck perspectives.
- Experiments on both  Frank Kitchen and MetaWorld demonstrate its superior performance.

**Weaknesses:**

- The baselines compared in this paper are not originally video-conditioned. DT is proposed to be conditioned on states, and VIMA is conditioned on texts and images. I encourage the authors to include a video-based baseline to strengthen this paper.

**Questions:**

- What is the generalization ability of your proposed method? Can it generalize to unseen tasks or unseen visual backgrounds? This is an important factor which should be investigated.

---

> ### Author Response · Authors · 2024-11-23
> **Official Comment to Reviewer DKw6 (1/1)**
>
> ## Weakness 1
> The baselines compared in this paper are not originally video-conditioned. DT is proposed to be conditioned on states, and VIMA is conditioned on texts and images. I encourage the authors to include a video-based baseline to strengthen this paper.
> ## Answer 1
> Thank you for your suggestion, a more competitive video-conditioned baseline would indeed strengthen our paper. We have added a new video-conditioned baseline: ViP[1]. Although the purpose of ViP is to learn a video-conditioned policy, it has a different setting from our method. Thus, we have made the following modifications to adapt it to our setting:
> - Since we do not consider human videos as input, we have removed the part that uses human videos for pre-training.
> - Since we do not assume access to video labels, we have changed its supervised contrastive learning part to unsupervised contrastive learning on robot videos.
>
> We used the same codebase as WL-DM to implement ViP with minimal modifications and conducted experiments in the MetaWorld environment. The experimental results are as follows. ViP outperformed other baselines in this environment, demonstrating its effectiveness as a video-conditioned policy. However, its performance still lags behind WL-DM, which further demonstrates the effectiveness of our method.
>
> |   Env  /  Method  |        WL-DM         |        V-BET      |        V-DT       |        VIMA       |        ViP        |
> |:-----------------:|:--------------------:|:-----------------:|:-----------------:|:-----------------:|:-----------------:|
> |      MetaWorld    |$\textbf{2.57}\pm0.90$|    $1.18\pm0.85$  |   $1.24\pm0.86$   |   $0.87\pm0.84$   |   $1.91\pm0.88$   |
>
> ## Question 1
> What is the generalization ability of your proposed method? Can it generalize to unseen tasks or unseen visual backgrounds? This is an important factor which should be investigated.
> ## Answer 2
> In terms of the generalization ability, as we mentioned in Lines 19-22, our method mainly focuses on compositional generalization, meaning that it can generalize to videos containing unseen task combinations. Generalization to unseen tasks or visual features is beyond the scope of this paper, but it is indeed a valuable research direction, which may be considered in our future work. Thank you for your suggestion.
>
> [1] Chane-Sane, Elliot, et al. "Learning video-conditioned policies for unseen manipulation tasks." 2023 IEEE International Conference on Robotics and Automation (ICRA). IEEE, 2023.

---

> > ### Comment · Reviewer_DKw6 · 2024-11-25
> >
> > Thanks for your responses to address my concerns. I decided to maintain my score considering the quality of this paper, other reviews, and the authors' responses. I tend to accept this paper because I believe the proposed method can further inspire future research in this direction.

---

> > > ### Author Response · Authors · 2024-11-25
> > > **Thanks for your response**
> > >
> > > Thank you for taking the time to review our paper and provide suggestions. Your feedback has greatly helped us improve our work.

---

### Official Review · Reviewer_zDKg · 2024-11-04

**Soundness:** 2
**Presentation:** 4
**Contribution:** 2
**Rating:** 6
**Confidence:** 4

**Summary:**

This paper presents Watch-Less-Do-More (WL-DM), an imitation learning framework for video-conditioned policy learning, to enable an agent to learn multiple skills from videos and generalize to unseen task combinations. The method uses an information bottleneck to implicitly discover skills and decompose video demonstrations into tasks without requiring explicit video segmentation annotations. WL-DM is evaluated in environments like Frank Kitchen and MetaWorld, demonstrating its capacity to achieve compositional generalization in unseen task combinations, outperforming baseline methods.

**Strengths:**

- The article is clearly written and proposes an effective solution to the problem of skill discovery without relying on language information or manual annotation.

**Weaknesses:**

1. There lack of comparative experiments on learning directly from the segmented sub-task videos, so it is hard to see whether the method achieves its intuition of *Focusing on the current task*.
2. In potential real-world applications, the various steps in multi-step tasks are often causally linked, and the appearance of the same task in different videos may be different. Additionally, the accessible training videos are not guaranteed to cover all elements in downstream tasks, without including elements out of a certain task set. As a result, the generalizability of the method appears to be somewhat limited for now.

**Questions:**

1. Is there any ablation about the number of videos in the training set? This includes how many types of task combinations there are, and how many videos with different initializations are there for each combination.
2. How similar do two video clips from different training videos need to be to be considered as the same task?

---

> ### Author Response · Authors · 2024-11-23
> **Official Comment to Reviewer zDKg (1/2)**
>
> ## Weakness 1
> There lack of comparative experiments on learning directly from the segmented sub-task videos, so it is hard to see whether the method achieves its intuition of Focusing on the current task.
> ## Answer 1
> Thanks for your suggestion. As we mentioned in lines 217-219 of our paper, obtaining video segmentation annotations at the sub-task level is often difficult in many cases. Therefore, in our work, we assume that such video segmentation annotations do not exist and thus did not train policy using segmented sub-task videos for comparison. Additionally, when dealing with videos that contain multiple sub-tasks, a policy trained on segmented sub-task videos would still require an additional mechanism or model to determine which sub-tasks are included in the video and when to switch between sub-tasks (like the method used in Vid2Robot[1]). Therefore, we did not include such a baseline for comparison.
>
> ## Weakness 2
> In potential real-world applications, the various steps in multi-step tasks are often causally linked, and the appearance of the same task in different videos may be different. Additionally, the accessible training videos are not guaranteed to cover all elements in downstream tasks, without including elements out of a certain task set. As a result, the generalizability of the method appears to be somewhat limited for now.
> ## Answer 2
>
>     In potential real-world applications, the various steps in multi-step tasks are often causally linked, and the appearance of the same task in different videos may be different.
>
> Yes, the situations you mentioned could indeed occur, and our experiments have already included such scenarios. For example, in Franka kitchen, if there is a task involving a kettle, we always pick up the kettle from one place and then put it down in another. In this case, it can be considered that there is a causal relationship between "picking up the kettle" and "putting down the kettle". Similarly, in Franka kitchen, once the light is turned on, it will remain on and continuously affect subsequent tasks, so there are indeed scenarios where the same task is performed with different scenes (whether or not a light has been turned on before performing the task). However, as shown in our experimental results, our algorithm still performs well in scenarios that include these situations.
>
>     Additionally, the accessible training videos are not guaranteed to cover all elements in downstream tasks, without including elements out of a certain task set.
>
> It is a common assumption in compositional generalization problems that the training set can cover all elements. Indeed, this assumption may not apply to all real-world scenarios, and thus compositional generalization cannot cover all generalization scenarios (compositional generalization is a subset of generalization problems). However, problems involving compositional generalization are not rare in practice. For example, when robots are applied to industrial assembly lines, they often encounter only limited types of components. In such cases, we can construct a training set that covers all types of components, which can then be used to train the robot policy. During execution, the robot policy that can achieve compositional generalization can then generalize to different combinations of components.
>
>     As a result, the generalizability of the method appears to be somewhat limited for now.
>
> In short, we agree with your point that the compositional generalization problem is a subset of the generalization problem. However, as mentioned above, the experiment scenarios considered in our paper already cover some of the situations you pointed out. Moreover, as shown in the above example, compositional generalization problems are not rare in real life. Therefore, we believe that our research on compositional generalization and the method proposed in this paper are meaningful to some extent.
>
> [1] Jain, Vidhi, et al. "Vid2robot: End-to-end video-conditioned policy learning with cross-attention transformers." arXiv preprint arXiv:2403.12943 (2024).

---

> ### Author Response · Authors · 2024-11-23
> **Official Comment to Reviewer zDKg (2/2)**
>
> ## Question 1
> Is there any ablation about the number of videos in the training set? This includes how many types of task combinations there are, and how many videos with different initializations are there for each combination.
> ## Answer 3
> Thank you for your suggestion. We have added the following ablation studies to further demonstrate our method.
>
> Firstly, we have included an experiment on the number of task combinations in the training set in MetaWorld environment. In our paper, we included 17 task combinations (7/3 split) in the training set, and here we further consider cases where we have 15 task combinations (6/4 split) and 20 task combinations in the training set.
>
> | Train-Test Split/ Method|           WL-DM         |           V-BET         |          V-DT           |           VIMA          |
> |:-----------------------:|:-----------------------:|:-----------------------:|:-----------------------:|:-----------------------:|
> |         6/4             | $\textbf{1.93}\pm0.26$  |       $1.34\pm0.78$     |      $0.86\pm0.84$      |      $0.43\pm0.78$      |
> |         7/3 (paper)     | $\textbf{2.57}\pm0.90$  |       $1.18\pm0.85$     |      $1.24\pm0.86$      |      $0.87\pm0.84$      |
> |         8/2             | $\textbf{1.88}\pm0.36$  |       $0.63\pm0.86$     |      $0.94\pm0.88$      |      $0.81\pm0.61$      |
>
> Secondly, we have included an experiment in MetaWorld environment on the number of videos corresponding to each task combination. In the paper, we considered 20 different videos for each task combination, and here we further consider cases with 40 different videos for each task combination.
>
> |      #Video/ Method     |           WL-DM         |           V-BET         |           V-DT          |           VIMA          |
> |:-----------------------:|:-----------------------:|:-----------------------:|:-----------------------:|:-----------------------:|
> |         20 (paper)      |  $\textbf{2.57}\pm0.90$ |       $1.18\pm0.85$     |      $1.24\pm0.86$      |      $0.87\pm0.84$      |
> |         40              |  $\textbf{2.21}\pm0.81$ |       $1.71\pm0.63$     |      $1.33\pm0.90$      |      $1.09\pm0.63$      |
>
> As shown in the above experimental results, WL-DM still outperforms other baselines, further demonstrating the effectiveness of our method.
>
> ## Question 2
> How similar do two video clips from different training videos need to be to be considered as the same task?
> ## Answer 4
> We used the expert policy to perform various tasks in the environment and collected corresponding videos. Therefore, whether two video clips are considered to be the same task depends on whether the expert policy executed the same task during video recording.

---

> ### Comment · Area_Chair_Cb37 · 2024-11-25
> **Please read rebuttal**
>
> Dear  Reviewer zDKg,
> Could you please read the authors' rebuttal and give them feedback at your earliest convenience? Thanks.
> AC

---

> ### Comment · Reviewer_zDKg · 2024-11-26
>
> Thanks for the authors' response. The rebuttal and further experiment results in the authors' replies to the reviewers enhance the paper. As a result, I have raised my score.

---

> > ### Author Response · Authors · 2024-11-26
> > **Thanks for your response**
> >
> > Thank you for taking the time to review our paper and provide suggestions. Your feedback has greatly helped us improve our work.

---

### Meta-Review · Area_Chair_Cb37 · 2024-12-17

**Metareview:**

This paper proposes an imitation learning framework that learns from videos to discover multiple skills. The proposed framework can enable robots to generalize to unseen combinations of tasks. We encourage the authors to revise the paper and add the experiments during rebuttal into the text. All the reviewers recognize the clarity of this paper.

**Additional Comments On Reviewer Discussion:**

The authors added new experiments with video-based baselines. Most of the reviewers are satisfied with the current empirical results.

The authors added text that explains the contribution of the method and the difference between the proposed approach and other mutual info-based approaches.

The authors provided a thorough explanation for potential real-world challenges.

---

### Decision · Program_Chairs · 2025-01-22

Accept (Poster)